# Structural basis of tubulin detyrosination by VASH2/SVBP heterodimer

Chen Zhou[1,2], Ling Yan[1,2], Wen-hui Zhang[1] & Zhu Liu [1]

The C-terminus of α-tubulin undergoes a detyrosination/tyrosination cycle and dysregulation of this cycle is associated with cancer and other diseases. The molecular mechanisms of tubulin tyrosination are well studied, however it has remained unknown how tyrosine is cleaved from the tubulin tail. Here, we report the crystal structure of the long-sought detyrosination enzyme, the VASH2/SVBP heterodimer at 2.2 Å resolution and the structure of the tail/VASH2/SVBP complex at 2.5 Å resolution. VASH2 possesses a non-canonical Cys-His-Ser catalytic architecture for tyrosine cleavage. The dynamics of the α1- and α2-helices of VASH2 are related to the insolubility of VASH2. SVBP plays a chaperone-like role by extensively interacting with VASH2 and stabilizing these dynamic helices. A positively charged groove around the catalytic pocket and the α1- and α2- helices of VASH2 targets the tubulin tail for detyrosination. We provide insights into the mechanisms underlying the cycle of tubulin tyrosine cleavage and religation.

[1] National Key Laboratory of Crop Genetic Improvement, College of Life Science and Technology, Huazhong Agricultural University, Wuhan 430070, China. [2] These authors contributed equally: Chen Zhou, Ling Yan. Correspondence and requests for materials should be addressed to Z.L. (email: liuzhu@mail.hzau. edu.cn)

Microtubules, as crucial components of the eukaryotic cytoskeleton, are involved in a variety of cellular functions. Microtubules are dynamically assembled from α- and β-tubulin heterodimers. A tubulin consists of a folded domain and an unstructured C-terminal tail. A large number of chemically diverse posttranslational modifications (PTMs) on the tubulin tail produce microtubule heterogeneity, leading to "tubulin code" patterns[1–4]. These encoded signals on the tubulin tail recruit motors and microtubule-associated proteins (MAPs) to endow microtubule with specialized functions[5–7]. Exploring the molecular mechanisms of these chemical reactions of PTMs on the tubulin tail is fundamental to unraveling tubulin code and to understanding microtubule functions[8–10].

Among these enzymatic posttranslational modifications, α-tubulin tyrosination is the first identified modification[11–13]. α-Tubulin encoded with a C-terminal tyrosine undergoes a detyrosination/tyrosination cycle in all cell types[14]. In this cycle, the C-terminal tyrosine is removed by tubulin carboxypeptidase (TCP) and can be religated by tubulin tyrosine ligase (TTL). This cycle regulates the abundance of detyrosinated/tyrosinated tubulin, and dysregulation is linked to diseases and cancers[15,16]. Physiologically, detyrosinated tubulin adapts the functions of microtubule by modulating their interacting proteins. For example, detyrosination prolongs microtubule lifetime up to 16 h[17] by protecting microtubule from depolymerizing motor proteins[18,19]. Stable detyrosinated microtubule buckle with desmin to contract cardiomyocytes, providing myocytes with mechanical resistance[20]. Detyrosinated microtubules orient towards the cell equator to guide CENP-E–dependent chromosome motion during mitosis[21]. Motor kinesin-1 preferentially binds to detyrosinated microtubules as a signal for recruiting and moving vimentin intermediate filaments onto microtubules[22,23], and interaction between kinesin-1 and detyrosinated microtubule navigates this motor into axons to participate in neuron development[24–28].

Although the physiological importance of α-tubulin detyrosination has been extensively studied, the long-sought carboxypeptidase that removes tyrosine from tubulin has only recently been identified[28,29]. It was found that a complex of vasohibins (VASHs) and vasohibin binding protein (SVBP) exerts a detyrosination function in which the VASHs digest the tubulin C-terminal tyrosine, while SVBP plays a chaperone-like role by increasing the solubility of the VASHs[28,29]. However, the molecular mechanism underlying this biology remains unknown.

Here we report the crystal structure of the VASH2/SVBP heterodimer at 2.2 Å resolution, as well as the crystal structure of the tubulin tail bound to this complex at 2.5 Å resolution. VASH2 possesses a non-canonical Cys-His-Ser catalytic architecture. In this catalytic triad, a water molecule is sandwiched between the serine side chain and the Nε2H of histidine, orienting the imidazole ring to form a thiolate/imidazolium ion pair for nucleophilic attack and tyrosine cleavage. The α1- and α2- helices of VASH2 are dynamic and the structural fluctuations of these helices are related to the insolubility of VASH2. SVBP plays a chaperone-like role by extensively interacting with VASH2 and stabilizing the α1- and α2- helices. A positively charged groove around the catalytic pocket on VASH2 is found to target the tubulin tail for detyrosination. The α1- and α2- helices of VASH2 are also required for tubulin tail detyrosination.

## Results

**Structure of VASH2/SVBP heterodimer.** VASH2 alone is poorly soluble[28–30]; here, we co-expressed full-length VASH2$_{1–355}$ and SVBP$_{1–66}$ from *Mus musculus* in *Sf9* insect cells (Fig. 1a; Supplementary Fig. 1). Highly-purified and well-behaved complexes were subjected to crystallization. The crystal structure of the

VASH2$_{1–355}$/SVBP$_{1–66}$ heterodimer was determined in the space group $C222_1$ at a resolution of 2.5 Å using iodide-based single-wavelength anomalous diffraction (Supplementary Fig. 2; Supplementary Table 1). Poor electron density of some residues of VASH2 (1–48 and 300–355) and some residues of SVBP (1–25 and 61–66) are unable to be generated a well-built model, indicating they are likely to be flexible. To improve the structure resolution, we prepared a complex of N-terminal truncated VASH2$_{47–355}$ and the full-length SVBP$_{1–66}$ for crystallization, and determined the crystal structure of the VASH2$_{47–355}$/SVBP$_{1–66}$ heterodimer at 2.2 Å resolution by molecular replacement using the VASH2$_{1–355}$/SVBP$_{1–66}$ structure as search model (Fig. 1b; Supplementary Table 1). These two structures are almost identical, with a root mean square deviation (RMSD) of 0.28 (Supplementary Fig. 3). Our analyses and discussion are based on the VASH2$_{47–355}$/SVBP$_{1–66}$ structure with higher resolution.

Two copies of the VASH2/SVBP heterodimer were observed in an asymmetric unit, and they are structurally identical with a RMSD of 0.16 Å (Supplementary Fig. 4). To clarify if the heterodimer can grow into higher order oligomers, we performed analytical ultracentrifugation analysis against a range of VASH2/SVBP complex concentrations. The sedimentation coefficient distribution ($c(s)$) of VASH2/SVBP under different protein concentration (20–70 μM) showed a single peak, and the experimentally measured molecular weight was almost equal to the sum of VASH2 and SVBP (Supplementary Fig. 5). Thus, the VASH2/SVBP heterodimer does not form higher order oligomeric structures under the condition analyzed. Possibilities of higher order oligomers of VASH2/SVBP under higher protein concentration cannot be exclude.

In the complex, SVBP folds as a single α-helix associating with VASH2, and VASH2 folds into two interacting domains of roughly equal length that form a V-shaped configuration (Fig. 1b). Previously, bioinformatics prediction and biochemical data have proposed VASH2 as a transglutaminase-like (GTL) cysteine protease possessing a non-canonical Cys-His-Ser catalytic triad[28,29,31]. The secondary structure and topology of VASH2 are similar to that of a GTL cysteine protease (Fig. 2a). However, their global folds are distinct[32,33] (Figs. 1b, 2a). The N-terminal domain (NTD) of VASH2 contains five successive α-helices (α1, α2, α3, α4 and α5), and the C-terminal domain (CTD) contains a β-sheet made of five antiparallel β-strands and two α-helices (with β1-β2-β3-α6-β4-β5-α7 linking topology). Unlike GTL protein folding, the α1- and α2-helices of VASH2 are twisted by the SVBP helix to form a helical bundle. The C-terminal α7-helix is displaced from the N-terminus by interacting with the α6-helix and β5-strand in CTD. Together, the non-canonical cysteine protease VASH2 presents a different conformation from the canonical GTL cysteine protease, suggesting a non-canonical catalytic architecture for nucleophilic attack (see below).

NTD and CTD of VASH2 delimit a cleft between them, where the Cys-His-Ser catalytic triad is formed at the interface of the α5-helix and two successive β strands (with C158 located at the tip of the α5-helix, H193 located on the β2-strand, and S210 located behind the β3-strand) (Fig. 1b). The architecture of this non-canonical Cys-His-Ser catalytic triad is different from that of the canonical Cys-His-Asp catalytic triad (Fig. 2c, right panel). In the Cys-His-Asp catalytic triad of the GTL cysteine protease, the imidazole ring of histidine is oriented to form a thiolate/imidazolium ion pair either by forming hydrogen bonds between the aspartic acid side chain and Nε2H of histidine or indirectly by water mediation, and eventually, an activated sulfur nucleophile is prepared for nucleophilic attack. In contrast, in the non-canonical Cys-His-Ser catalytic triad of VASH2, the distance between the serine side chain and the Nε2H of histidine is extended to 4.3 Å, so that the imidazole ring would not be well oriented for

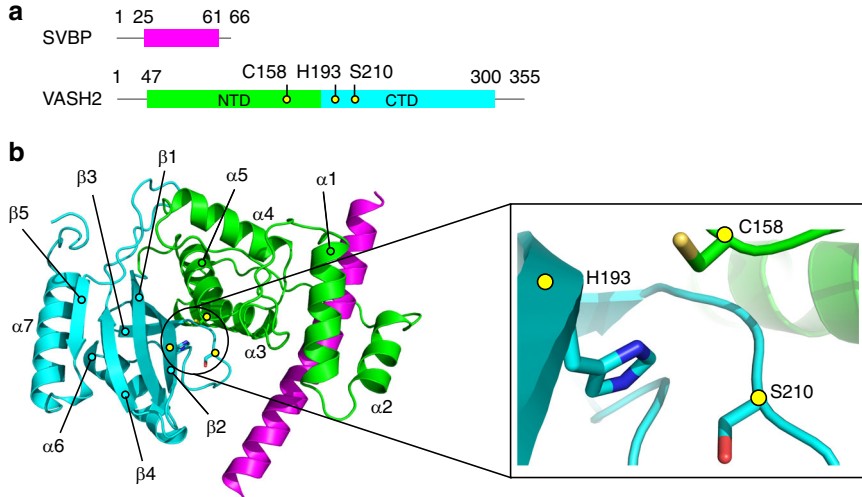

**Fig. 1** Structure of the VASH2/SVBP heterodimer. **a** Schematic depiction of VASH2 and SVBP. The Cys-His-Ser catalytic triad is highlighted in yellow circles. **b** Cartoon representation of the VASH2/SVBP structure. VASH2 NTD is colored green, CTD is colored cyan, and SVBP is colored magenta. C158, H193 and S210 are shown in stick representation. The α-helices and β-strands are numbered and highlighted

thiolate/imidazolium ion pair formation by serine/histidine interaction. Instead, the imidazole ring is oriented indirectly via a water molecule, by twisting the ring to rotate a degree of 45.6.

**SVBP stabilizes VASH2 by extensive interactions**. The structure of VASH2/SVBP heterodimer allows us to understand how SVBP stabilizes VASH2 by playing a chaperone-like role. SVBP was reported to associate with VASH2 to enhance VASH2 solubility, which is essential for functioning[28,29]. Our structure reveals extensive interactions between SVBP and VASH2. Approximately 1314 Å² of the SVBP accessible surface is buried by VASH2, which is 36% of the SVBP surface participating in the intermolecular interactions (with the total accessible surface of SVBP being 3652 Å²) (Fig. 3a). These interactions are derived from a series of polar interactions between the SVBP helix and coupled residues in the VASH2 NTD helical bundle and CTD loops (Supplementary Table 2). Owing to these strong interactions, alanine scanning of interacting residues on SVBP or VASH2 is unable to disrupt the complex. Only multiple alanine mutations on either or both proteins or charge reversal mutations can disrupt the complex, as revealed by the significant attenuation of soluble VASH2 species during VASH2/SVBP co-expression and purification (Supplementary Fig. 6). These data indicate that SVBP association and particular interacting residues are essential for VASH2 solubility.

To gain more insights into the mechanism that how SVBP enhances VASH2 solubility, we performed molecular dynamics simulations of VASH2 in the absence or presence of SVBP. The simulations results revealed that the backbone RMSD fluctuation of VASH2 without SVBP is approximately 2–3 times that of VASH2 in the presence of SVBP (Fig. 3b). Specifically, the fluctuating regions of VASH2 are located at the α1- and α2-helices (excluding some terminal and inner loops), and the two helices fluctuate integrally and maintain their helical fold during simulations (Fig. 3c; Supplementary Figs. 7 and 8). Whereas, SVBP significantly reduces these fluctuations (Fig. 3c). Thus, we suspected that the dynamics of the two helices would make VASH2 insoluble. In other words, VASH2 would be soluble without these helices. Then, we designed a VASH2 construct without α1- and α2-helices (by 1–93 truncation), named VASH2_Δα1/α2. Unlike the full-length VASH2, the VASH2_Δα1/α2 is soluble and shows well

behavior on gel filtration column (Supplementary Fig. 9). Collectively, the structural changes and flexibility of the α1- and α2-helices are related to VASH2 insolubility, and mechanistically, SVBP makes VASH2 soluble by stabilizing these helical fluctuations.

**Recognition and detyrosination of tubulin tail by VASH2**. By analyzing the electrostatic surface of the VASH2/SVBP heterodimer, we found that there is a continuous positively charged groove surrounding the Cys-His-Ser catalytic triad pocket (Fig. 4a; Supplementary Fig. 10). The positive potential on the complex surface is mainly derived from a series of basic residues in VASH2, including R134, K135, K157, K183, R211, R212, K218, K237, K238, K244, K245, K247, K265 and K273. This is probably a region to recognize the negatively charged α-tubulin tail and to facilitate tyrosine cleavage in the catalytic pocket (the tail comprises the VDSVEGEGEEEGEEY sequence). To assess the roles of this positively charged surface in tubulin detyrosination, we used a reversed-phase high-performance liquid chromatography (HPLC) assay to analyze the mutational impacts of these abovementioned residues on tyrosine cleavage (Fig. 4b). First, we confirmed that VASH2/SVBP can efficiently cleave the C-terminal tyrosine from the tubulin tail. Accordingly, a catalytically dead mutant, VASH2(C158A)/SVBP, displays no detyrosination activity, and other mutations, including H193A and S210A in the catalytic triad, exhibit dramatically reduced detyrosination efficiency (Fig. 4b, c).

Then, we generated a series of charge reversal mutations in the positively charged groove on VASH2, and quantified the detyrosination efficiency changes caused by these mutations (Fig. 4c; Supplementary Fig. 10). Some mutations lead to VASH2 insolubility (including K237E, K244E, K245E, K265E and K273E), and other available VASH2/SVBP mutants all attenuate detyrosination efficiency to various degrees. Some charge reversal mutations near the Cys-His-Ser catalytic triad pocket, including K135E, K157E, R211E, R212E and K218E, dramatically reduce detyrosination efficiency. In contrast, other mutations far from the pocket, including R134E, K183E, K238E and K247E, do not markedly reduce detyrosination efficiency. As a control, a charge reversal mutation on the back side of the positive surface (K119E) had no impact on detyrosination (Fig. 4c; Supplementary Fig. 10). Notably, mutations to the two consecutive residues R134 and

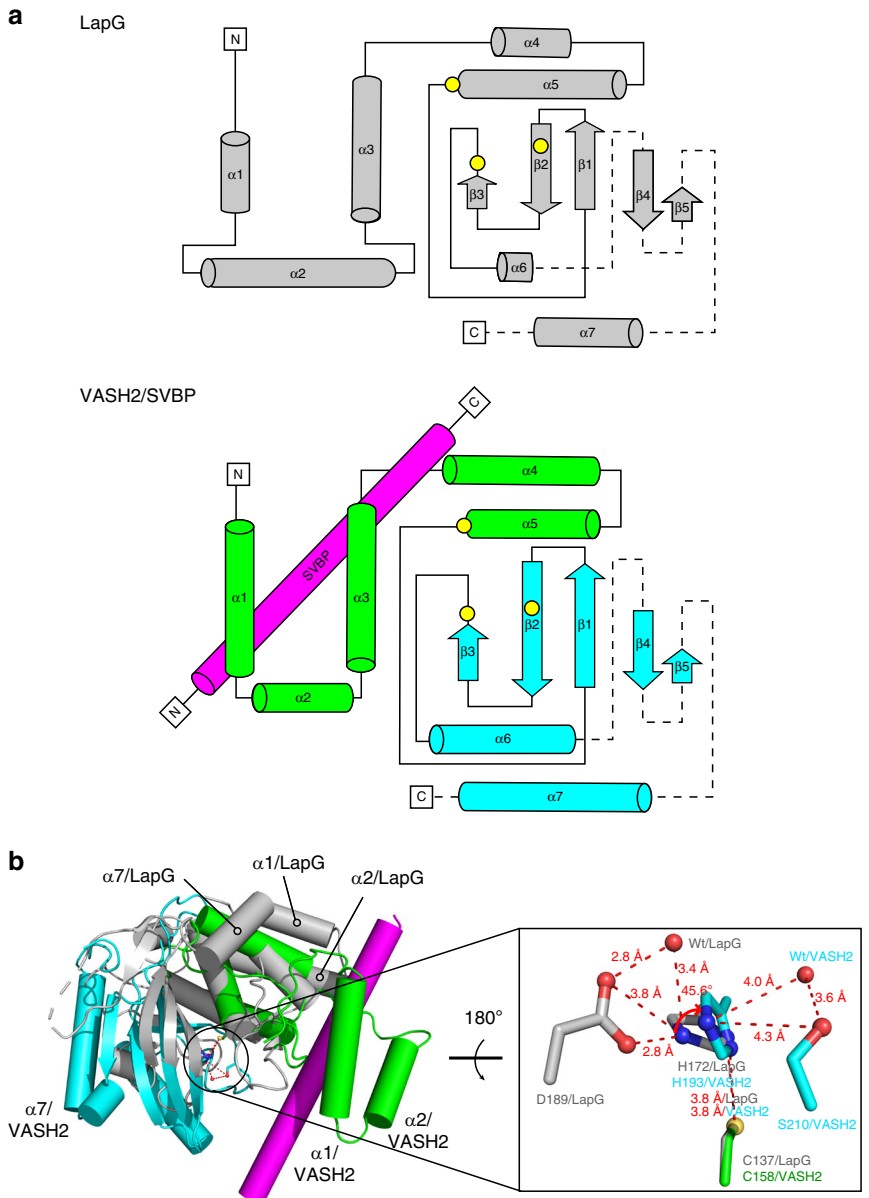

**Fig. 2** Structural comparison between LapG and VASH2. **a** Secondary structure and topological diagrams of the two cysteine proteases. LapG is colored gray, NTD and CTD of VASH2 are colored green and cyan, and SVBP is colored magenta, respectively. The residues positions of the catalytic triad are highlighted in yellow circles. **b** LapG (PDB ID: 4FGP) is aligned to VASH2 using the catalytic cysteine (C137 in LapG and C158 in VASH2). Structures are colored in the same scheme in (**a**). The pocket of the catalytic triad is zoomed-in with residues shown in stick representation. Two water molecules in the pocket are shown as red spheres. The red arrow indicates a 45.7° rotation of the imidazole ring

K135 produce opposite effects on detyrosination (Fig. 4c). The R134E mutation slightly reduces detyrosination efficiency, whereas the K135E mutation almost inhibits tyrosine cleavage. This result can be attributed to the fact that the positively charged side chain of K135 points to the catalytic pocket, which may be conducive to negatively charged tail recognition and tyrosine cleavage. However, the side chain of R134 points out of the pocket and may not be involved in the tail recognition (Supplementary Fig. 11). Ultimately, we mapped the key residues on VASH2 for tubulin tail detyrosination, and showed that the electrostatic force between the positively charged groove near the catalytic pocket and the glutamate-rich tubulin tail is crucial for tyrosine cleavage. Other positively charged surfaces outside the catalytic pocket may be involved in interacting with negatively charged tubulin folded domain[8,9].

Furthermore, we analyzed the roles of the α1- and α2-helix of VASH2 in tubulin tail detyrosination. We showed that VASH2 is soluble and well behaved without its dynamic helices and SVBP (Supplementary Fig. 9). Then, we checked whether this truncated VASH2 could function. The detyrosination efficiency of VASH2_Δα1/α2 is significantly reduced to 20.6%, and adding of SVBP into VASH2_Δα1/α2 (at a 1:1 mole ratio) produces little enhancement (Fig. 4c). Alone, SVBP has no detyrosination activity (Fig. 4c). Thus, the α1- and α2- helices are also required for tubulin tail detyrosination, and the stabilized helices by SVBP may be positioning the tail for hydrolysis in the catalytic pocket and/or by inducing the correct conformation in VASH2.

To directly visualize tubulin tail recognition by VASH2/SVBP, we tried to determine the structure of the tail/VASH2/SVBP complex. By co-crystallizing VASH2$_{1-355}$/SVBP$_{1-66}$ with the tail

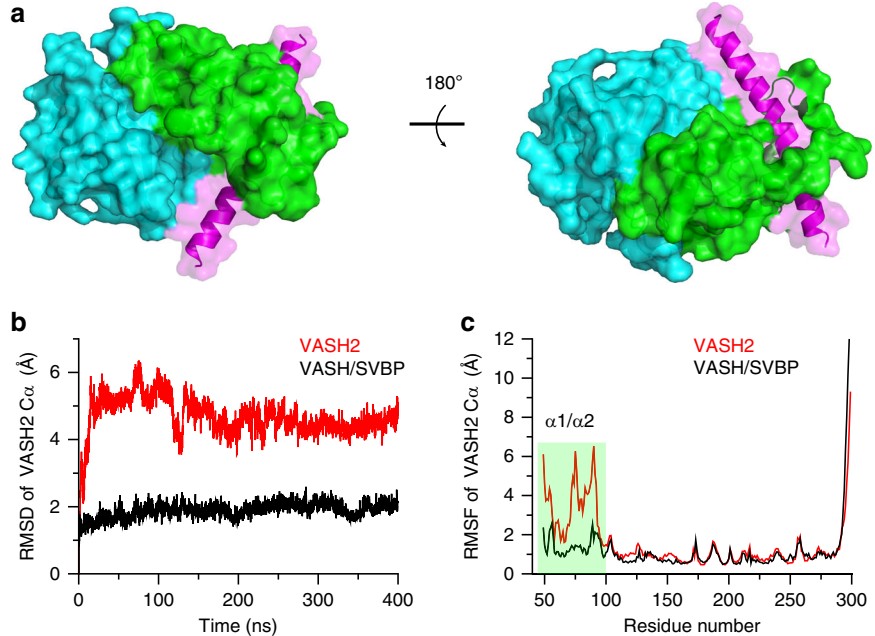

**Fig. 3** Dynamics of VASH2 and stabilization by extensive interaction with SVBP. **a** Surface representation of contacts between VASH2 and SVBP. **b** Representative RMSD fluctuations of VASH2 C$_\alpha$ atoms in the MD trajectory. The fluctuation of VASH2 in the presence and in the absence of SVBP is represented by black lines and red lines, respectively. **c** The root mean square fluctuation (RMSF) of VASH2 C$_\alpha$ atoms during the simulations is calculated with reference to an energy-minimized initial crystal structure. Green shading highlights the α1- and α2- helices of VASH2 possessing high RMSF without SVBP

of different residue lengths, only co-crystallization with the full-length tail (VDSVEGEGEEEGEEY) gave rise to crystals and exhibited good diffraction. Using the VASH2$_{1-355}$/SVBP$_{1-66}$ structure as the search model, we determined the ternary structure of tail/VASH2/SVBP at a resolution of 2.5 Å by molecular replacement, in which the last five residues of tubulin tail (EGEEY) were built near the catalytic pocket of VASH2. (Fig. 4d; Supplementary Fig. 12; Supplementary Table 1). The structure of VASH2/SVBP in the ternary complex is identical to the *apo* form with an RMSD of 0.30 Å (Supplementary Fig. 13a). The negatively charged tail interacts with some VASH2 residues in the positively charged groove and in the catalytic pocket, including R211, R212, K157, and A158 (A158 is mutated from C158 to block detyrosination in order to enable enzyme/substrate crystallization) (Fig. 4d). This finding is consistent with our activity analysis, which showed that mutations of these residues significantly reduce detyrosination efficiency (Fig. 4c). Since the distance between the nucleophilic atom and the tyrosine peptide bond exceeds 9 Å, a nucleophilic attack is not allowed. Thus, the tail conformation captured in VASH2/SVBP is in a pre-arranged state, and the tail is about to be pulled deeper into the catalytic pocket for tyrosine cleavage (Fig. 4d and Supplementary Fig. 13b). Some other efforts to stabilize the tubulin full-length tail in the catalytic pocket are further needed to see how the tyrosine is cleavage by the cysteine.

## Discussion

We have determined the structure of the VASH2/SVBP heterodimer and depicted the structural basis for α-tubulin detyrosination. Compared to the canonical GTL cysteine protease, VASH2 protease adopts a different conformation and possesses a non-canonical Cys-His-Ser catalytic architecture. In the catalytic triad, a His-water-Ser hydrogen bond network is necessary for thiolate/imidazolium ion pair formation. SVBP plays a chaperone-like role by enhancing VASH2 solubility[28,29], and our data revealed that the α1- and α2- helices of VASH2 are dynamic

and the structural fluctuation and flexibility of these helices are related to the insolubility of VASH2. Our data revealed that SVBP plays a chaperone-like role by extensively interacting with VASH2 and stabilizing the α1- and α2- helices of VASH2.

Based on structural and mutational analysis, we revealed that a positively charged groove around the catalytic pocket on VASH2 is crucial for tubulin tail detyrosination. We mapped key residues for tyrosine cleavage on VASH2 and found that basic residues near the catalytic pocket are essential for tail recognition and detyrosination, while other basic residues far from the pocket have fewer roles and may be responsible for the recognition of the negatively charged tubulin folded domain[8,9]. Furthermore, we found that the α1- and α2-helices are also necessary for VASH2 full activity. Collectively, our results revealed the very important roles of these helices, that they not only relate to VASH2 solubility, but also contribute to tyrosine cleavage.

We captured a pre-arranged state of the tubulin tail in the catalytic pocket of VASH2, and this conformation of tail/VASH2/SVBP is convinced by our charge reversal mutation analysis. We propose that the positive groove would target tubulin for detyrosination (Supplementary Fig. 13c). In addition to TTL and TTL-like family[4–6], a typical positively charged groove should be a common feature of PTM enzymes for tubulin modifications. Overall, we provide insights into the mechanisms underlying the cycle of tubulin tyrosine cleavage and religation. Further structural studies of microtubules/VASH2/SVBP complex are required to reveal the preference of VASHs for microtubules over α- and β-tubulin heterodimers[28].

## Methods

**Protein preparation**. The genes *Vasohibin-2 (VASH2)* (accession number NM_144879) and *SVBP* (accession number NM_024462) were amplified from the *Mus musculus* complementary DNA library using a standard PCR-based protocol, subcloned into a modified pFastBac1 vector and verified by DNA sequencing. All primer sequences for PCR are listed in the Supplementary Table 3. SVBP was fused with a His-tag at the N-terminus, whereas VASH2 was fused with no tag. The protein complex of VASH2$_{1-355}$/SVBP$_{1-66}$ (full-length proteins),

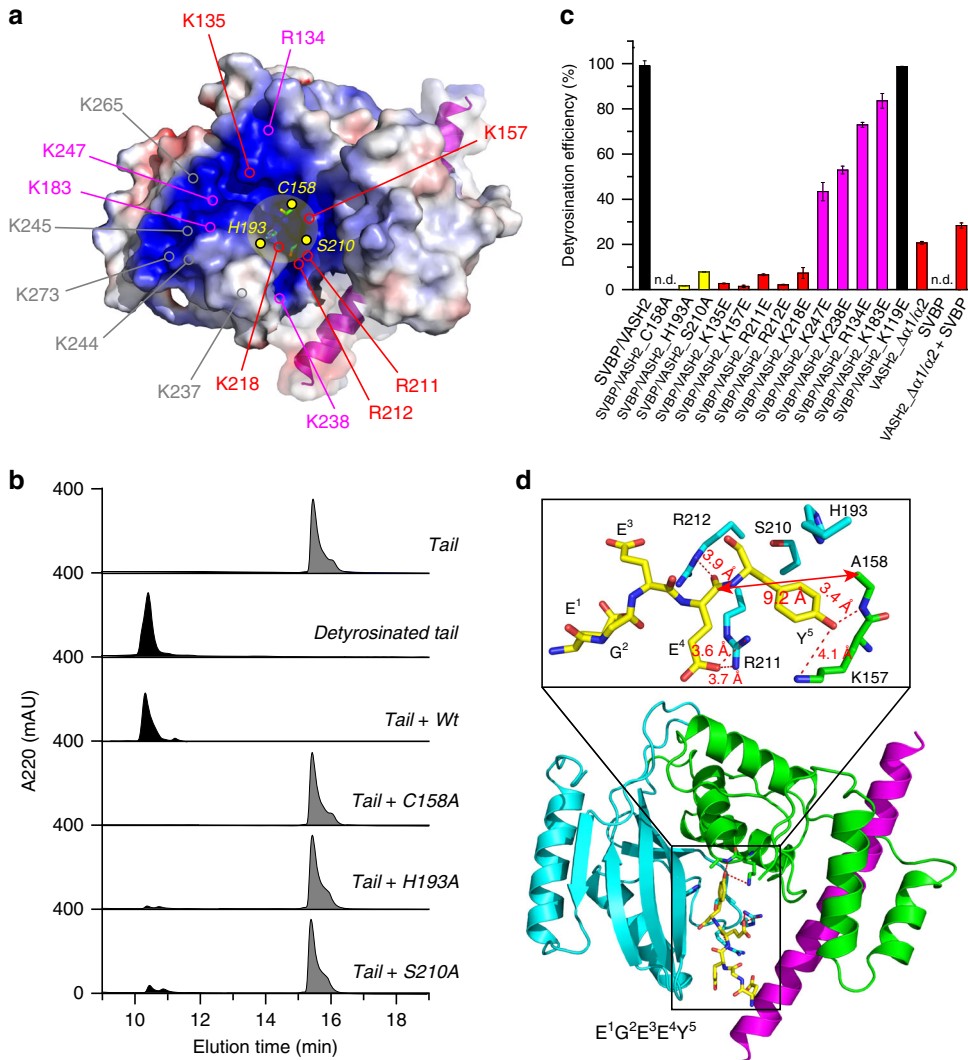

**Fig. 4** Tubulin tail recognition and detyrosination by VASH2/SVBP. **a** Electrostatic surface of VASH2/SVBP is colored in terms of electrostatic potential, displayed in a scale from red (−5 kT/e) to blue (+5 kT/e). The N- and C-terminal loops of VASH2 are omitted for better visualization. SVBP is shown in both cartoon and surface representations, and its surface transparency is set to 0.4. The residues of the catalytic triad are highlighted in yellow and are shown in stick representation. Yellow shading denotes the catalytic pocket. Positively charged residues contributing to the potential on VASH2 are also highlighted. **b** HPLC characterizes the detyrosination efficiency of VASH2/SVBP and its mutants. Gray peaks correspond to the tubulin tail (VDSVEGEGEEEGEEY), and black peaks correspond to detyrosinated tail (VDSVEGEGEEEGEE). The percentages of black peak areas represent the detyrosination efficiency of the enzyme. **c** Detyrosination efficiency of VASH2/SVBP, mutants, and SVBP. The data from three independent measurements are averaged, and the error indicates SD. **d** The structure of VASH2/SVBP binds with a five-residue peptide. VASH2/SVBP is shown in cartoon representation, whereas peptide and interacting residues in VASH2 are shown in stick representation. The zoomed-in box in the up panel illustrates the detailed interactions. Raw data of (**b**) and (**c**) are provided as a Source Data file

VASH2₄₇₋₃₅₅/SVBP₁₋₆₆, and relevant mutants (including VASH2₁₋₃₅₅(C158A)/SVBP₁₋₆₆ and other mutants used in Fig. 4c) was co-expressed in *Spodoptera frugiperda Sf*9 cells (Gibco, 12659017), respectively. Cells were cultured at 27 °C for 60 h after co-infected separate baculoviruses using the Bac-to-Bac system(Life-Technologies). Cells were harvested by centrifugation at 1500 × g for 20 min and homogenized in ice-cold lysis buffer containing 25 mM Tris-HCl, pH 8.0, 200 mM NaCl and 1 mM phenylmethanesulfonyl-fluoride (PMSF). The cells were disrupted using a cell homogenizer, and the insoluble fraction was precipitated by ultra-centrifugation (14,000 × g) for 1 h at 4 °C. The supernatant was loaded onto a Ni-NTA superflow affinity column, washed with lysis buffer plus 5 mM imidazole, and then eluted with elution buffer containing 25 mM Tris-HCl, pH 8.0, 200 mM NaCl and 250 mM imidazole. The targets were then further polished using a heparin sepharose column (GE Healthcare) by eluting under a linear gradient of NaCl concentration between 0–1 M in a buffer containing 25 mM Tris-HCl, 1 mM dithiothreitol (DTT). The His-tag was removed by incubation with DrICE at 4 °C for 3 h and then subjected to size-exclusion chromatography (SEC 650, Bio-Rad) in a buffer containing 25 mM Tris-HCl, pH 8.0, 200 mM NaCl and 5 mM DTT. The peak fractions of targets conformed by Tricine-SDS-PAGE and were collected for crystallization and biochemical assays.

VASH2_α1/α2 (94–355) was cloned into a modified pFastBac1 vector with a C-terminal Flag tag. Protein was also expressed using *Spodoptera frugiperda Sf*9 cells. Cells were cultured at 27 °C for 60 h after baculovirus infection, harvested and homogenized in lysis buffer containing 25 mM Tris-HCl, pH 8.0, 200 mM NaCl. After ultracentrifugation, the supernatant was loaded onto a Flag-affinity column, washed using lysis buffer and eluted using a buffer containing 25 mM Tris-HCl, pH 8.0, 200 mM NaCl, 0.3 mg/ml Flag peptide. The target was then subjected to a heparin sepharose column and finally polished using SEC 650 in a buffer containing 25 mM Tris-HCl, pH 8.0, 200 mM NaCl and 5 mM DTT. The Flag tagged VASH2_α1/α2 (94–355) was used for detyrosination activity assay without Flag cleavage.

SVBP₁₋₆₆ was cloned into a modified pFastBac1 vector with a N-terminal His-tag and was also expressed using *Spodoptera frugiperda Sf*9 cells. Cells were cultured at 27 °C for 60 h after baculovirus infection, harvested and homogenized in lysis buffer containing 25 mM Tris-HCl, pH 8.0, 200 mM NaCl, 1 mM PMSF. After ultracentrifugation, the supernatant was loaded onto a Ni-NTA superflow affinity column, washed using lysis buffer plus 5 mM imidzole and eluted using a buffer containing 25 mM Tris-HCl, pH 8.0, 200 mM NaCl, 250 mM imidzole. The target was then subjected to a heparin sepharose column and finally polished using SEC 650 in a buffer containing 25 mM Tris-HCl, pH 8.0, 200 mM NaCl and 5 mM DTT.

**Crystallization**. Crystallization was performed using the sitting-drop vapour diffusion method at 18 °C by mixing equal volumes (1 μl) of protein (5 mg ml$^{-1}$) and reservoir solution. VASH2$_{1-355}$/SVBP$_{1-66}$ crystals were grown in drops containing 0.1 M HEPES pH 7.0, 1 M (NH$_4$)$_2$SO$_4$, and 1.125 M KCl. High-quality VASH2$_{1-355}$/SVBP$_{1-66}$ crystals were soaked in the reservoir solution plus 0.25 M 5-amino-2,4,6-triiodoisophthalic acid (I3C, Molecular Dimensions) for 5–10 min. The crystals acquired a pale yellow color within 5–10 min and were immediately immersed in cryoprotectant solution. VASH2$_{47-355}$/SVBP$_{1-66}$ (7 mg ml$^{-1}$) crystals were grown in drops containing 0.1 M BIS-TRIS propane pH 7.0 and 2.0 M Sodium formate. Plate-shaped crystals appeared within 2–3 days and grew to full size within 7–10 days. To obtain crystals of a ternary complex of tail/ VASH2$_{1-355}$/ SVBP$_{1-66}$, different length of the α-tubulin tail peptides (1 mM) were co-crystallized with VASH2$_{1-355}$(C158A)/SVBP$_{1-66}$ (5 mg ml$^{-1}$). After numerous trials, only the full-length tail (VDSVEGEGEEGEEY) in complex with VASH2$_{1-355(C158A)}$/SVBP$_{1-66}$ gave rise to crystals in 0.1 M HEPES, pH 7.0, 1 M (NH$_4$)$_2$SO$_4$ and 1.105 M KCl that exhibited good diffraction. Cryo protection of the VASH2$_{47-355}$/SVBP$_{1-66}$ crystals is 0.1 M BIS-TRIS propane pH 7.0, 2.0 M Sodium formate and 25 % Glycerol. Cryo protection of the VASH2$_{1-355}$/SVBP$_{1-66}$ crystals soaked with I3C is Santovac® Cryo Oil (Hampton Research). Cryo protection of the VASH2$_{1-355}$(C158A)/SVBP$_{1-66}$ crystals co-crystallized with α-tubulin tail peptides is 0.1 M HEPES, pH 7.0, 1 M (NH4)$_2$SO$_4$, 1.105 M KCl and 25 % Glycerol.

**Data collection and structure determination**. All data sets were collected at the SSRF beamlines BL17U1 and BL19U1[34] and were processed with the HKL2000 program and XDS packages[35]. Further data processing was carried out using the CCP4 program suite[36]. The data collection and structure refinement statistics are summarized in Supplementary Table 1. The structure of the VASH2$_{1-355}$/SVBP$_{1-66}$ complex was solved by the single-wavelength anomalous diffraction (SAD) method with I3C-soaked crystals. The program SHELX with HKL2MAP graphic interface[37] was used for the initial phase determination and the generation of initial electron density maps. Twelve heavy atom signals of iodine were observed in the asymmetric unit. The initial model was built de novo by COOT[38]. The structure of VASH2$_{47-355}$/SVBP$_{1-66}$ complex and peptide-bound VASH2$_{1-355}$(C158A)/ SVBP$_{1-66}$ complex were solved by molecular replacement using the I3C-bound structure as the searching model with PHASER[39]. The structures were built using COOT and iteratively refined using the PHENIX program[40]. Figures were generated using PyMOL (http://www.pymol.org/).

**Analytical ultracentrifugation**. Analytical ultracentrifugation (AUC) experiments were performed using ProteomeLab XL-I analytical ultracentrifuge equipped with An-50 Ti rotor (Beckman Coulter). VASH2$_{1-355}$/SVBP$_{1-66}$ was prepared in a buffer of 25 mM Tris-HCl, pH 8.0 and 200 mM NaCl. Sample was loaded in 12-mm double-sector aluminium centerpieces (Beckman Coulter) and ran at 147,420 g. Data were collected via absorbance detection at 18 °C. Absorbance of 20, 40 and 70 μM VASH2$_{1-355}$/SVBP$_{1-66}$ complex was collected at 280, 250 and 215 nm, respectively. SV-AUC data were globally analyzed using the SEDFIT program and fitted to a continuous $c(s)$ distribution model to determine the molecular mass. Data analysis was carried out using OriginPro 8.

**Molecular dynamics simulations**. All MD simulations of VASH2$_{47-355}$ as well as the complex with SVBP$_{1-66}$ were performed using AMBER 16 software[41] with ff14SB force field[42]. The starting conformation is our crystal structure. The proteins were solvated in a cube containing TIP3P water[43], with at least 10 Å padding in all directions. Ions, including sodium ion and chloride ion, were also added to neutralize the electric charge from the proteins. Long-range electrostatics were treated the with partial mesh ewald (PME)[44] method, and van der Waals interactions were truncated at 10 Å with an energy shift. The time step was 2 fs, and the SHAKE[45] algorithm was used to constrain the bonds connecting hydrogen atoms. The entire system was first energy minimized and heated to 298 K before the production process. The trajectory production processes proceeded for 400 ns to obtain 4000 snapshots at 100 ps intervals. The RMSD and RMSF of the C$_\alpha$ atoms were calculated using the PTRAJ module in the AMBER package.

**Characterization of the interaction between VASH2 and SVBP**. SVBP is needed for the solubility of VASHs[28–30]. Therefore, a co-expression and purification strategy was used to characterize the interaction between VASH2$_{1-355}$ and SVBP$_{1-66}$ (see Supplementary Fig. 5). In this expression system, the cDNAs for the VASH2$_{1-355}$ mutant and SVBP$_{1-66}$ were cloned into the viral expression vector pFastBacDual (Invitrogen), to enable the sequential sub-cloning of the two foreign genes into two separate cassettes for co-expression. SVBP$_{1-66}$ was fused with a His-tag at the N-terminus, whereas VASH2$_{1-355}$ was fused with no tag. The protein complex was co-expressed in *Spodoptera frugiperda Sf9* cells at 27 °C for 60 h by baculovirus infection using the Bac-to-Bac system. The complexes were purified by Ni-NTA superflow affinity column and were eluted by 20 mM Tris-HCl, pH 8.0, 200 mM NaCl, 250 mM imidazole. Eluent were detected by 16% Tricine-SDS-PAGE gels.

**Measurement of detyrosination efficiency**. The detyrosination assay with an α–tubulin C-terminal tail peptide exploits the decreased hydrophobicity of the detyrosinated (product) peptide relative to the substrate to separate them through reversed-phase high-performance liquid chromatography (HPLC). The synthetic peptide-substrate (VDSVEGEGEEGEEY) and peptide-product (VDSVEGEGEEE-GEE) (Biosynthesis) were dissolved in 25 mM Tris-HCl, pH 8.0, 200 mM NaCl. Detyrosination of the peptide-substrate was performed in a 50 μl reaction system consisting of 25 mM Tris-HCl, pH 8.0, 200 mM NaCl, 1 mM peptide-substrate, and 10 μM wildtype VASH2$_{1-355}$/SVBP$_{1-66}$ or mutated complex at room temperature for 45 min. The reaction was quenched by 17.5% acetonitrile and 0.05% trifluoroacetic acid, and the sample was subjected to HPLC(LC-20A&SOLUTION) using a reversed-phase (RP) analytical column (Vydac, 218MS C18 5 mm, 25 × 0.46 cm). The peptides were separated by RP HPLC with an acetonitrile gradient (10–35% in 30 min) in 0.05% trifluoroacetic acid as the eluent. The quantities of both the substrate and product peptides were estimated from the areas of the corresponding peaks on the UV detection absorbance spectrum (220 nm), and the percentages of product formation represent the detyrosination efficiency.

**Reporting Summary**. Further information on research design is available in the Nature Research Reporting Summary linked to this article.

## Data Availability
Atomic coordinates of the VASH2/SVBP heterodimer, I3C-bound heterodimer, and tail-bound form have been deposited in the Protein Data Bank (PDB) under accession number 6JZC, 6JZE, and 6JZD, respectively. The source data underlying Fig. 4b, c are provided as a Source Data file. Other data are available from the corresponding author upon reasonable request.

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

## Acknowledgements
We thank staffs at the BL17U1, BL19U1 and BL19U2 beamline of the NCPSS at Shanghai Synchrotron Radiation Facility for assistance with data collection, and research associate Dr. De-lin Zhang at the Center for Protein Research, Huazhong Agricultural University, for facilities support. We thank Dr. Zhou. Gong at the Wuhan Institute of Physics and Mathematics, CAS, for MD data analysis. The work has been supported by the National Key R&D Program of China (2018YFA0507700 to Z.L.), and by the Fundamental Research Funds for the Central Universities (Program No. 2662019PY004 to Z.L.).

## Author contributions
Z.L. and C.Z. designed the project and analyzed data, C.Z. and W.H.Z prepared protein sample, C.Z. performed crystallization and biochemical assay, and L.Y. resolved the structure. Z.L. performer MD simulations and wrote the manuscript with support from all authors.

## Additional information

**Competing interests:** The authors declare no competing interests.

