## [Peer Review File · Nature Communications]

Reviewers' comments:

Reviewer #1 (Remarks to the Author):

In this manuscript authors study the detyrosination mechanism of α -tubulin through the carboxypeptidase complex VASH/SVPB. This is an important post-translational modification within tubulin heterodimer that affects microtubules lifetime and stability providing myocytes with mechanical resistance, among other functions. The manuscript show very interesting results within the field but I do not see strong evidences for its conclusions. Instead, further experiments will be needed. Also, I am not a native speaker but I found difficult to follow some paragraphs and many misspelled words. I would recommend text editing.

Authors have solved 3 crystal structures related to the complex VASH/SVBP; one soaked with I3C for SAD experiments, one to get high resolution and one soaked with a peptide related to α -tubulin C-terminal tail. They found 2 heterodimers in the a.s.u in one of the structures and perform AUC experiments to clarify the presence of higher order oligomerization structures. Based also on their structure they perform further experiments to: i) determine the importance of SVBP on VASH stabilization (by MD and single and multiple Ala mutagenesis follow by co-purification), ii) determine the α -tubulin tail binding site (co-crystallization) and, iii) determine the relevance of a positively charged groove on detyrosination efficiency (charge-reversal mutation and HPLC). I have some concerns related to all three-crystal structures and the AUC experiment. Also, authors do not mention that SVBP used in their experiments is proteolysed (as you can see from SDS-PAGE gels of purification of co-expressed mutants). Specific comments are below

Related to the manuscript I have the following concerns

1. AUC experiments. According to previous reports VASH2 cannot be purified in the absence of SVBP, but using only 1 concentration is not enough to clarify if the heterodimer can grow into higher order oligomers as you suspect from the presence of 2 heterodimers within the asu (You do not show the arrangement of these 'oligomer' and so, it is difficult to predict its likelihood). In case you want to determine if there is any chance VASH2/SVBP oligomerization into higher order structures the classical AUC speed experiment includes a range of protein concentrations. Ideally, you should also try each protein alone and titration of one protein with the other. However, with your co-expression system you are forcing a 1:1 SVBP:VASH2 ratio
I also found 56000rpm a little bit high for such molecular weight (46-48KDa), but not sure about this point because you do not mention the rotor you use.

2. You purify both proteins from the His Tag linked to SVBP and obtain the heterodimer after a final gel filtration step. You say that SVBP is key for VASH2 solubility through the stabilization of VASH2 α 1 and α 2 helices that are unstable, but actually none of your experiments discern if SVBP is required as a chaperone or as a co-factor driving the correct conformation for catalysis or as a co-factor positioning α -tubulin tail for catalysis. MD results show α 1 and α 2 movement, which could be related to protein instability (as you mention) but also to a conformational change or higher flexibility at this region, instead of instability. Also most of the basic residues within the positive groove required for α -tubulin tail interaction are not in SVBP-VASH2 interface. Considering your results I think that you should try a construct of VASH2 without α 1 and α 2, which should be stable and would not require SVBP for purification. Since these helices do not contain the catalytic triad, the truncated protein should also be able to function without SVBP. If not, then SVBP is required for α -tubulin tail positioning upon hydrolysis and/or induce the correct conformation in VASH2.

3. you say "... determine the structure via SAD method with I3C and to refine the complex structure to 2.2A. Two copies of VASH2/SVBP heterodimer were observed in the asu...". According to your data you solved phases with I3C in a crystal where there was 1 heterodimer in the asu and THEN solved a second crystal by MR where there were 2 heterodimers in the asu due to an increase of the unit cell on c. It is interesting how soaking experiments (also happen with the peptide) reduce the unit cell making only 1 heterodimer in asu instead of 2. Is the crystal packing conserved?

4. Structures: data collection statistics table:

a. please split between data collection statistics and refinement statistics

- b. please give the value of CChalf for each of the structures solved
- c. please give information about SAD experiment: anomalous correlation, anomalous redundancy, wavelength, number of sites ...

5. Structures: comments:

- a. Table S1 show very similar Ramachandran plot statistics for all 3 structures with a 0.4% of not allow, whereas in the validation report there is only 1 residue out in 6JZD. Why is that discrepancy?
 - b. 6JZC: the validation report highlights several outliers (bond lengths, chiral, side chains) and many clashes (clashes due to symmetry molecules, solvent GOL clashes with protein chains, etc). This is quite annoying considering 2.20Å resolution. I think you should take care of these problems before final deposition in PDB
 - c. 6JZE: is the structure solved by SAD and there is no information related to this experiment. Validation report highlight that 3 of 4 I3C molecules have an occupancy less than 0.9 (which is the real occupancy of these molecules?, how are the statistics for the anomalous signal? Also the average B factors of other entities (in this structure this is related to I3C) are really high, which make me suspect about the real position and signal of these molecules.
 - d. 6JZD: why are there chain A, B and D? where is chain C? chain B (α -tubulin tail peptide) contains a 60% of outliers. This chain contains only 5 residues ordered, which ring the bell about the accuracy of you building process. Also average B factors of other entities are really high and there are very few water molecules considering your resolution range. I think this structure needs further inspection before data deposition.
6. You mention that VASH2 CTD consists of 5 stranded antiparallel β -sheets. I guess you mean that consists on a β -sheet made of 5 antiparallel β -strands

7. Extra density on 6JZC. You do not mention how your peptide in 6JZD fit into this extra density to support this could be a cell natural substrate. In figure S9, do you show the composite omip map? or this is the map after the MR?. Considering this is a supplemental figure I will ask for a bigger figure to clearly see what you are mentioning within the text

8. Which is the rmsd of the alignment between VASH2 and GTL?

9. Figure 2. It is impossible to see the localization of the labelled residues on the structure even highlighting them on sticks representation. This figure needs a zoom into the area you want to highlight. Red and green colours are very bad chooses since colour blind people cannot see anything. Yellow is impossible to distinguish on a white background.

10. Figure S6. I can identify 2 bands at HisSVBP level that suggest protein proteolysis. Which version of the protein binds to VASH2? Also HisSVBP mutant R34E/R36E runs as a higher molecular weight protein, how can be this possible? This sample shows a lot of contamination. Have you do mass spectrometry of all your constructs?

Reviewer #2 (Remarks to the Author):

The manuscript of Zhou et al. describes work to investigate the molecular structure of VASH2/SVBP and the structural basis for its interaction the C-terminal tail of tubulin, which it detyrosinates. This type of information about tubulin post-translational modification mechanisms is only beginning to accumulate, so way(s) in which the 'tubulin code' is written is still poorly understood. The authors present a crystal structure of an apo VASH2/SVBP heterodimer, and another VASH2/SVBP heterodimer complexed to a short tubulin tail peptide. The structure shows the extensive interactions between VASH2 and SVBP, and helps explain the stability SVBP imparts to the bi-lobed structure of VASH2. On the opposite face, a large positive charge-rich cavity surrounds the unusual catalytic pocket, where tyrosine is cleaved away. The authors provide some biochemical data to identify residues involved in this reaction, and those that could play supporting roles, such as guiding the tubulin tail toward the active site. Residues with little or no role, yet in the same vicinity, are identified too. The ternary structure of the peptide + VASH2/SVBP heterodimer complex supports some of this data and shows the orientation of tyrosine on

approach to the catalytic pocket.

Overall the study is interesting and important, but the manuscript requires quite a bit of editing to clean up the writing and grammar. There are also several places (in the main text and in figure legends) where the authors seem to be referring to the stability of VASH2 when bound to SVBP, but they replace SVBP with VASH2.

Reviewer #1 (Remarks to the Author):

In this manuscript authors study the de tyrosination mechanism of α -tubulin through the carboxypeptidase complex VASH/SVPB. This is an important post-translational modification within tubulin heterodimer that affects microtubules lifetime and stability providing myocytes with mechanical resistance, among other functions. The manuscript show very interesting results within the field but I do not see strong evidences for its conclusions. Instead, further experiments will be needed. Also, I am not a native speaker but I found difficult to follow some paragraphs and many misspelled words. I would recommend text editing.

Thank you for your recognition of the importance of our work, and we appreciate these very insightful comments. According to these constructive suggestions, we have performed more experiments and analyses. These results were included in the revised manuscript. We have carefully organized and proofread this manuscript. Our revised manuscript has also been edited by Nature Research Editing Service and it should be most free of syntax errors.

Authors have solved 3 crystal structures related to the complex VASH/SVBP; one soaked with I3C for SAD experiments, one to get high resolution and one soaked with a peptide related to α -tubulin C-terminal tail. They found 2 heterodimers in the a.s.u in one of the structures and perform AUC experiments to clarify the presence of higher order oligomerization structures. Based also on their structure they perform further experiments to: i) determine the importance of SVBP on VASH stabilization (by MD and single and multiple Ala mutagenesis follow by co-purification), ii) determine the α -tubulin tail binding site (co-crystalization) and, iii) determine the relevance of a positively charged groove on de tyrosination efficiency (charge-reversal mutation and HPLC). I have some concerns related to all three-crystal structures and the AUC experiment. Also, authors do not mention that SVBP used in their experiments is proteolysed (as you can see from SDS-PAGE gels of purification of co-expressed mutants). Specific comments are below

Related to the manuscript I have the following concerns

1. AUC experiments. According to previous reports VASH2 cannot be purified in the absence of SVBP, but using only 1 concentration is not enough to clarify if the heterodimer can grow into higher order oligomers as you suspect from the presence of 2 heterodimers within the asu (You do not show the arrangement of these 'oligomer' and so, it is difficult to predict its likelihood). In case you want to determine if there is any chance VASH2/SVBP oligomerization into higher order structures the classical AUC speed experiment includes a range of protein concentrations. Ideally, you should also try each protein alone and titration of one protein with the other. However, with your co-expression system you are forcing a 1:1 SVBP:VASH2 ratio

I also found 56000rpm a little bit high for such molecular weight (46-48KDa), but not sure about this point because you do not mention the rotor you use.

Thank you for your very insightful comments. The AUC data was collected by using ProteomeLab XL-I analytical ultracentrifuge equipped with An-50 Ti rotor (Beckman Coulter), and sample was loaded in 12-mm double-sector aluminium centerpieces (Beckman Coulter). We have updated this information in Methods of the revised manuscript ("Analytical ultracentrifugation (AUC)" in page 17).

According to your constructive suggestions, we have collected more AUC data of 20 μ M, 40 μ M and 70 μ M VASH2/SVBP using a lower rotor speed of 45000 rpm. It turned out that the sedimentation coefficient distribution showed a single peak regardless of the protein concentration, and the experimentally measured molecular weight was almost equal to the sum of VASH2 and SVBP (Supplementary Fig. 5 in the revised manuscript). Thus, VASH2/SVBP heterodimer is monomeric in solution under these sub-micromolar protein concentrations. However, possibilities of higher order oligomers of VASH2/SVBP under much higher protein concentration cannot be exclude. We have included these data and comments in the revised manuscript (paragraph 1 in page 6).

2. You purify both proteins from the His Tag linked to SVBP and obtain the

heterodimer after a final gel filtration step. You say that SVBP is key for VASH2 solubility through the stabilization of VASH2 $\alpha 1$ and $\alpha 2$ helices that are unstable, but actually none of your experiments discern if SVBP is required as a chaperone or as a co-factor driving the correct conformation for catalysis or as a co-factor positioning α -tubulin tail for catalysis. MD results show $\alpha 1$ and $\alpha 2$ movement, which could be related to protein instability (as you mention) but also to a conformational change or higher flexibility at this region, instead of instability. Also most of the basic residues within the positive groove required for α -tubulin tail interaction are not in SVBP-VASH2 interface.

Considering your results I think that you should try a construct of VASH2 without $\alpha 1$ and $\alpha 2$, which should be stable and would not require SVBP for purification. Since these helices do not contain the catalytic triad, the truncated protein should also be able to function without SVBP. If not, then SVBP is required for α -tubulin tail positioning upon hydrolysis and/or induce the correct conformation in VASH2.

Thank you for your very constructive suggestions.

Firstly, in revision, we designed a VASH2 construct without $\alpha 1$ - and $\alpha 2$ -helix (by 1-93 truncation), named VASH2_ $\Delta\alpha 1/\alpha 2$. Unlike full-length VASH2, the truncated VASH2 is soluble and well behaved on gel filtration column (Supplementary Fig. 9 in the revised manuscript). Thus, the $\alpha 1$ - and $\alpha 2$ - helices of VASH2 are related to VASH2 insolubility, and that appears to be generated by the dynamics of these helices.

Secondly, thank you for your very insightful comments for the "instability" and "flexibility" of these helices. We have deep-analyzed our MD results and found that these helices fluctuated integrally and maintained their helical fold during MD simulations, instead of unfolding (we added a figure as Supplementary Fig. 8 in the revised manuscript). Thus, in the revised manuscript, we used "structural changes and flexibility of $\alpha 1$ - and $\alpha 2$ - helices" to replace "stability of $\alpha 1$ - and $\alpha 2$ - helices" to describe their relevance in VASH2 solubility, and proposed that the SVBP favors VASH2 solubility by stabilizing these dynamic $\alpha 1$ - and $\alpha 2$ - helices (Abstract and

paragraph 1 in page 9).

Thirdly, we have measured the dephosphorylation activity of VASH2_Δ α 1/ α 2. It turned out that the activity of the truncated VASH2 was significantly reduced, and adding of SVBP into VASH2_Δ α 1/ α 2 produces little enhancements. Thus, the two helices are also necessary for VASH2 function, and as you say, they may be positioning tubulin tail for hydrolysis in catalytic pocket and/or by inducing correct conformation in VASH2. We included these results and your insightful comments in the revised manuscript (Fig 4c and paragraph 1 in page 11).

3. you say “... determine the structure via SAD method with I3C and to refine the complex structure to 2.2Å. Two copies of VASH2/SVBP heterodimer were observed in the asu...”. According to your data you solved phases with I3C in a crystal where there was 1 heterodimer in the asu and THEN solved a second crystal by MR where there were 2 heterodimers in the asu due to an increase of the unit cell on c. It is interesting how soaking experiments (also happen with the peptide) reduce the unit cell making only 1 heterodimer in asu instead of 2. Is the crystal packing conserved?

We are sorry for the misleading. In the revised manuscript, we have described more details for the sample preparation and structure determination.

Firstly, we determined the I3C soaked structure and the *apo* structure using below method (paragraph 1 in page 5 in the revised manuscript):

"The crystal structure of the VASH2₁₋₃₅₅/SVBP₁₋₆₆ heterodimer was determined in the space group $C222_1$ at a resolution of 2.5 Å using iodide-based single-wavelength anomalous diffraction (Supplementary Fig. 2; Supplementary Table 1). Poor electron density of some residues of VASH2 (1-48 and 300-355) and some residues of SVBP (1-25 and 61-66) are unable to be generated a well-built model, indicating they are likely to be flexible. To improve the structure resolution, we prepared a complex of N-terminal truncated VASH2₄₇₋₃₅₅ and the full-length SVBP₁₋₆₆ for crystallization, and determined the crystal structure of the VASH2₄₇₋₃₅₅/SVBP₁₋₆₆ heterodimer at 2.2 Å resolution by molecular replacement using the VASH2₁₋₃₅₅/SVBP₁₋₆₆ structure as

search model (Fig. 1b; Supplementary Table 1). These two structures are almost identical, with a root mean square deviation (RMSD) of 0.28 (Supplementary Fig. 3). Our analyses and discussion are based on the VASH2₄₇₋₃₅₅/SVBP₁₋₆₆ structure with higher resolution."

Secondly, we determined the ternary structure of tubulin tail/VASH2/SVBP using below method (paragraph 2 in page 11 in the revised manuscript):

"To directly visualize tubulin tail recognition by VASH2/SVBP, we tried to determine the structure of the tail/VASH2/SVBP complex. By co-crystallizing VASH2₁₋₃₅₅/SVBP₁₋₆₆ with the tail of different residue lengths, only co-crystallization with the full-length tail (VDSVEGEGEEEGEEY) gave rise to crystals and exhibited good diffraction. Using the VASH2₁₋₃₅₅/SVBP₁₋₆₆ structure as the search model, we determined the ternary structure of tail/VASH2/SVBP at a resolution of 2.5 Å by molecular replacement, in which the last five residues of tubulin tail (EGEEY) were built near the catalytic pocket of VASH2. (Fig. 4d; Supplementary Fig. 12; Supplementary Table 1). The structure of VASH2/SVBP in the ternary complex is identical to the *apo* form with an RMSD of 0.30 Å (Supplementary Fig. 13a)."

Collectively, these differences in asymmetric unit should be related to the complex with different protein boundaries in crystallization. We are sorry for this missing information. More details have also been included in Methods of the revised manuscript ("Protein preparation", "Crystallization", and "Data collection and structure determination" in page 14-16).

4. Structures: data collection statistics table:

a. please split between data collection statistics and refinement statistics

b. please give the value of CChalf for each of the structures solved

c. please give information about SAD experiment: anomalous correlation, anomalous redundancy, wavelength, number of sites ...

These points are well-taken. We have updated the data collection and refinement

statistics table (Supplementary Table 1).

5. *Structures: comments:*

We appreciate your very insightful comments and constructive suggestions about our structures. We have carefully improved and re-described the 6JZC and 6JZD structures in the revised manuscript. The improved structures have been re-deposited to the wwPDB, and validated by wwPDB staff. Minor structural changes have no effect on our conclusions and we have re-drawn some relatively structural figures in the revised manuscript (including Fig. 2b, 4d; Supplementary Fig. 12, 13b). Below please find our point-by-point responses.

a. Table S1 show very similar Ramachandran plot statistics for all 3 structures with a 0.4% of not allow, whereas in the validation report there is only 1 residue out in 6JZD. Why is that discrepancy?

The reason for the discrepancy is that we used the PROCHECK program (Laskowski et al., 1993) to statistic the Ramachandran plot. In revision, we have refined our structure, avoiding outlier of Ramachandran. We have updated the data collection and refinement statistics table (Supplementary Table 1).

b. 6JZC: the validation report highlights several outliers (bond lengths, chiral, side chains) and many clashes (clashes due to symmetry molecules, solvent GOL clashes with protein chains, etc). This is quite annoying considering 2.20Å resolution. I think you should take care of these problems before final deposition in PDB.

We have carefully improved the structure and the coordinate file. We have re-deposited the improved structure to wwPDB and the validation report shows that the all-atom clash score is 3.

c. 6JZE: is the structure solved by SAD and there is no information related to this experiment. Validation report highlight that 3 of 4 I3C molecules have an occupancy less than 0.9 (which is the real occupancy of these molecules?, how are the statistics for the anomalous signal? Also the average B factors of other entities (in this

structure this is related to I3C) are really high, which make me suspect about the real position and signal of these molecules.

We are truly sorry for missing the information. 6JZE is the structure solved by SAD and we have included this information in the main text (paragraph 1 in page 5) and Methods ("Data collection and structure determination" in page 16) in the revised manuscript. The statistics for the anomalous signal have been updated in the Supplementary table 1.

Normally, since I3C was soaked into the crystals, the occupancy of I3C in crystals may be less than 1. In contrast, by reasonably reducing the occupancy during structure refinement, the B factor for I atom and average B factor for all atoms would be improved. Meanwhile, the density would be fitted better with the refined model. This strategy has been widely used to solve protein structures using anomalous signal of I atom. For example, in a *Nature communications* paper (2018, 9(1):1621), eight of nine I3C was given an occupancy less than 0.7 to refine a final model with a B factor of 57.3 for I atom and an average B factor of 49.0 for all atoms (occupancy for each I3C is 0.55, 0.40, 0.50, 0.64, 0.45, 0.46, 0.57, 0.45 and 1, respectively).

During the refinement of 6JZE structure, we constantly modified the occupancy of I3C and refined the model using PHENIX.REFINE; meanwhile, we traced the reducing of B factor of I3C and checked whether I3C was fitted into the density, to examine that the occupancy of I3C was reasonable. Ultimately, the appropriate occupancy of four I3C molecules is 0.84, 0.50, 0.74 and 0.76, respectively. After the refinement, and the B-factor of I atom was reduced from 180.8 to 148.5, and the average B factor for all atoms was 52.1. Moreover, we could see clearly iodide density in the 2Fo-Fc map contoured at 3σ (Supplementary Fig. 2), indicating that iodide atoms are at their real positions.

d. 6JZD: why are there chain A, B and D? where is chain C? chain B (α -tubulin tail peptide) contains a 60% of outliers. This chain contains only 5 residues ordered, which ring the bell about the accuracy of your building process. Also average B

factors of other entities are really high and there are very few water molecules considering your resolution range. I think this structure needs further inspection before data deposition.

Thank you for your very insightful comments. We have carefully improved this structure.

Firstly, we are sorry for the misleading and we have re-organized these chain IDs. In the updated coordinate file, VASH2, SVBP and tubulin tail is assigned with chain A, chain B and chain C, respectively.

Secondly, it is difficult to capture the final conformation of tubulin tail in hydrolyzing process. Although we used the tubulin full-length tail (VDSVEGEGEEEGEEY) for co-crystallization, only the last five residues (EGEEY) could be fitted into the refined model, indicating that the tail is likely to be flexible. In revision, we have carefully improved this structure and the outliers of E¹G²E³E⁴Y⁵ was reduced to 40% (E¹ and Y⁵ are the outliers). Admittedly, the B factors of these five residues were high. But the 2Fo-Fc map of peptide contoured at 1 σ showed that the density was fitted well with the refined atomic model (Supplementary Fig. 12 in the revised manuscript). In the structure, the tail locates at the catalytic area and shows the orientation of tyrosine on approach to the catalytic pocket. Thus, we called this conformation as "pre-arranged state". Some strategies to stabilize the tail in the pocket are further needed to see how the tyrosine is cleavage by the cysteine. Although it is challenging, we will try our best in the future. We have written this perspective in the revised manuscript (paragraph in page 12). Furthermore, The 4th Asp of tail (E¹G²E³E⁴Y⁵) is coordinated by R211 and R212 of VASH2, and the terminal tyrosine (Y⁵) interacts with the K157 of VASH2 (Fig.4 d in the revised manuscript). Our biochemical assay corroborated these observations that none of K157E, R211E or R212E exhibits comparable detyrosination activity to that of wild type (Fig. 4c in the revised manuscript).

Thirdly, water molecules were first automatically added using PHENIX program and then they were carefully checked to exclude unreasonable water molecules. In the

final model there were 16 water molecules at 2.5 Å resolution.

Overall, the improved structure is convinced and was re-deposited it to PDB.

6. You mention that VASH2 CTD consists of 5 stranded antiparallel β -sheets. I guess you mean that consists on a β -sheet made of 5 antiparallel β -strands

Thank you for you suggestions, we made it clear by using "C-terminal domain (CTD) contains a β -sheet made of five antiparallel β -strands".

7. Extra density on 6JZC. You do not mention how your peptide in 6JZD fit into this extra density to support this could be a cell natural substrate. In figure S9, do you show the composite omit map? or this is the map after the MR?. Considering this is a supplemental figure I will ask for a bigger figure to clearly see what you are mentioning within the text.

We appreciate this very insightful comments.

Firstly, according to your suggestions, we added the peptide (E¹G²E³E⁴Y⁵) to the same position in 6JZC and refined it. It turned out that the model did not fit well into composite omit map (Figure 1 below). We could not identify this molecule in the pocket even by collecting more than a hundred crystals of VASH2/SVBP for GC-MS analysis. Thus, it is hard to say that this density in 6JZC is a cell natural substrate, and it is needed much more efforts to identify. In the revised manuscript, we have removed the proposal that the unknown density may a cell nature substrate. Thank you for your comments.

Secondly, in figure S9b, that was a composite omit map after structure refinement and the 2Fo-Fc density was countered at 1 σ . We have presented it more clear in a bigger figure. Please see Supplementary Fig. 12 in the revised manuscript.

Figure 1. The 1σ $2F_o-F_c$ density map in the pocket of 6JZC.

8. Which is the rmsd of the alignment between VASH2 and GTL?

The structures of VASH2 and GTL cysteine protease are distinctly different. Their overall structures have almost no similarities (>10 Å). We have described the difference in Fig. 2b and in main text in the revised manuscript (paragraph 2 in page 6). To compare the catalytic architectures of the Cys-His-Asp triad of GTL and the Cys-His-Ser triad of VASH2, we aligned the two structures using their catalytic cysteine for discussion (Fig. 2b).

9. Figure 2. It is impossible to see the localization of the labelled residues on the structure even highlighting them on sticks representation. This figure needs a zoom into the area you want to highlight. Red and green colours are very bad choices since colour blind people cannot see anything. Yellow is impossible to distinguish on a white background.

Thank you for these constructive suggestions. We have re-drawn our figures in the revised manuscript (including Fig. 1b, Fig. 2b, Fig.4a-d, Supplementary Fig11, Fig13b).

10. Figure S6. I can identify 2 bands at HisSVBP level that suggest protein proteolysis. Which version of the protein binds to VASH2? Also HisSVBP mutant R34E/R36E runs as a higher molecular weight protein, how can be this possible? This sample shows a lot of contamination. Have you do mass spectrometry of all your constructs?

Thank you for your very insightful comments.

Firstly, in our previous manuscript, we ran the gels using Glycine-SDS-PAGE. In the revised manuscript, we have reran the gels using Tricine-SDS-PAGE. You will see that both VASH2 and HisSVBP showed a single band, and the HisSVBP mutant R34E/R36E ran normally (Supplementary Fig. 6 in revised manuscript). Thus, these proteolysis-like bands in our previous gels should be resulted from the buffer condition for gel running, instead of proteolysis. We have updated this figure (Supplementary Fig. 6).

Secondly, we used a co-expression and purification strategy to characterize the interaction between VASH2 and SVBP. The eluent of co-expressed complex in the first purification step (without further purification by heparin ion exchange and gel filtration) were subjected to SDS-PAGE analysis. Thus, there were some contamination (Supplementary Fig. 6). We have written more detailed methods in Methods of the revised manuscript ("Characterize the interaction between VASH2 and SVBP" in page 18).

Thirdly, in revision, we have checked two of our samples using ESI mass spectrometry (Bruker Daltonics). We measured molecular weights of the eluent in the first purification step of co-expressed wild type VASH2/SVBP (Figure 2a and 2b below) and co-expressed VASH2_R212A/SVBP_R34E/R36E (Figure 2c below). The measured molecular weights of VASH2, HisSVBP and HisSVBP mutant are almost equal to the theoretically calculated weights.

Collectively, we believe that these characterized interactions between VASH2 and SVBP are real and proteolysis during protein preparation should be most not happened.

Figure 2. ESI mass analysis of protein molecular weight.

Reviewer #2 (Remarks to the Author):

The manuscript of Zhou et al. describes work to investigate the molecular structure of VASH2/SVBP and the structural basis for its interaction the C-terminal tail of tubulin, which it detyrosinates. This type of information about tubulin post-translational modification mechanisms is only beginning to accumulate, so way(s) in which the 'tubulin code' is written is still poorly understood. The authors present a crystal structure of an apo VASH2/SVBP heterodimer, and another VASH2/SVBP heterodimer complexed to a short tubulin tail peptide. The structure shows the extensive interactions between VASH2 and SVBP, and helps explain the stability SVBP imparts to the bi-lobed structure of VASH2. On the opposite face, a large positive charge-rich cavity surrounds the unusual catalytic pocket, where tyrosine is cleaved away. The authors provide some biochemical data to identify residues involved in this reaction, and those that could play supporting roles, such as guiding the tubulin tail toward the active site. Residues with little or no role, yet in the same vicinity, are identified too. The ternary structure of the peptide + VASH2/SVBP heterodimer complex supports some of this data and shows the orientation of tyrosine on approach to the catalytic pocket.

Overall the study is interesting and important, but the manuscript requires quite a bit of editing to clean up the writing and grammar. There are also several places (in the main text and in figure legends) where the authors seem to be referring to the stability of VASH2 when bound to SVBP, but they replace SVBP with VASH2.

Thank you for your recognition of the importance of our work, and we appreciate your very insightful comments. According to these constructive suggestions, we have carefully organized and proofread this manuscript. Our revised manuscript has also been edited by Nature Research Editing Service and it should be most free of syntax errors.

REVIEWERS' COMMENTS:

Reviewer #1 (Remarks to the Author):

Authors have gone carefully through all my previous concerns and I think the manuscript has significantly improved. The new PDB validation reports support the improvement on the quality of the structures and authors have included missing information on the Crystallographic table.

I only have some minor revision.

Page 6: `... VASH2/SVBP heterodimer is monomeric in solution under these sub-micromolar protein concentration`.

You are using microM concentrations, no sub-microM. Instead, I would say; ` VASH2/SVBP heterodimer does not form higher order oligomeric structures under the condition analyzed`

Page 6: `... the a1 and a2 helices of VASH2 are twisted by the SVBP helix to form a helical bundle with a3 and a4 helices in VASH2-NTD`

I can see that a1 and a2 are twisted by SVBP when comparing to LapG structure and I see a1 and a2 forms a bundle, but I do not see that a3 and a4 as part of the bundle. Here, I do not understand what do you mean with `bundle`

Page 10: `Notably, mutations of the two consecutive residues R134 and R135`. It should stay K135.

Page 11: `...co-crystallizing VASH21-355/SVBP1-66 with the tail...`

But later you mention that actually this is the mutant VASH21-355(C158A)/SVBP1-66. You should also correct this on the crystallography table and on methods `data collection determination`

Page 14: `The protein complex of VASH21-355/SVBP1-66 (full-length proteins), VASH247-355/SVBP1-66, and relevant mutant was co-expressed ...`

Which mutant are you talking about? VASH21-355(C158A)/SVBP1-66?

Page 15: VASH2_a1/a2 (94-355).

I miss the expression conditions and purification buffers.

Page 15: His tagged VASH21-66

I guess you mean SVBP1-66. Also I miss the expression conditions and how it was prepared.

Page 16: AUC `Absorbance of ...was collected at 280 nm, 250 nm and 250 nm`

I guess last absorbance value is 220 or 215 nm

Reviewer #1 (Remarks to the Author):

Authors have gone carefully through all my previous concerns and I think the manuscript has significantly improved. The new PDB validation reports support the improvement on the quality of the structures and authors have included missing information on the Crystallographic table.

I only have some minor revision.

Thank you very much for your constructive suggestions to improve our manuscript.

Below please find our point-by-point responses about your comments.

Page 6: ‘... VASH2/SVBP heterodimer is monomeric in solution under these sub-micromolar protein concentration’.

You are using microM concentrations, no sub-microM. Instead, I would say; ‘VASH2/SVBP heterodimer does not form higher order oligomeric structures under the condition analyzed’

We appreciate this insightful comment and we have used this statement in the revised manuscript.

Page 6: ‘... the $\alpha 1$ and $\alpha 2$ helices of VASH2 are twisted by the SVBP helix to form a helical bundle with $\alpha 3$ and $\alpha 4$ helices in VASH2-NTD’

I can see that $\alpha 1$ and $\alpha 2$ are twisted by SVBP when comparing to LapG structure and I see $\alpha 1$ and $\alpha 2$ forms a bundle, but I do not see that $\alpha 3$ and $\alpha 4$ as part of the bundle. Here, I do not understand what do you mean with ‘bundle’

Thank you for your very insightful comment. The $\alpha 3$ - and $\alpha 4$ -helices do not form a helical bundle with $\alpha 1$ - and $\alpha 2$ -helices. We have deleted “**with the $\alpha 3$ - and $\alpha 4$ -helices in VASH2 NTD**” from “Unlike GTL protein folding, the $\alpha 1$ - and $\alpha 2$ -helices of VASH2 are twisted by the SVBP helix to form a helical bundle **with the $\alpha 3$ - and $\alpha 4$ -helices in VASH2 NTD**” in the revised manuscript.

Page 10: ‘Notably, mutations of the two consecutive residues R134 and R135’. It

should stay K135.

Thank you for pointing out this mistake. It has now been corrected in the revised manuscript.

Page 11: ‘...co-crystallizing VASH21-355/SVBP1-66 with the tail...’

But later you mention that actually this is the mutant VASH21-355(C158A)/SVBP1-66. You should also correct this on the crystallography table and on methods ‘data collection determination’

Thank you for your insightful comment. We have included this information into the crystallography table and methods “data collection and structure determination” in the revised manuscript.

Page 14: ‘The protein complex of VASH21-355/SVBP1-66 (full-length proteins), VASH247-355/SVBP1-66, and relevant mutant was co-expressed ...’

Which mutant are you talking about? VASH21-355(C158A)/SVBP1-66?

We are sorry for missing the information. Relevant mutants including VASH2₁₋₃₅₅(C158A)/SVBP₁₋₆₆ and other mutants used in Fig. 4c. We have included this information in the revised manuscript.

Page 15: VASH2_α1/α2 (94-355).

I miss the expression conditions and purification buffers.

Thank you for your suggestion. We prepared this sample using below method:

“VASH2_α1/α2 (94-355) was cloned into a modified pFastBac1 vector with a C-terminal Flag tag. Protein was also expressed using *Spodoptera frugiperda Sf9* cells. Cells were cultured at 27 °C for 60 h after baculovirus infection, harvested and homogenized in lysis buffer containing 25 mM Tris-HCl, pH 8.0, 200 mM NaCl. After ultracentrifugation, the supernatant was loaded onto a Flag-affinity column, washed using lysis buffer and eluted using a buffer containing 25 mM Tris-HCl, pH 8.0, 200 mM NaCl, 0.3 mg/ml Flag peptide. The target was then subjected to a

heparin sepharose column and finally polished using SEC 650 in a buffer containing 25 mM Tris-HCl, pH 8.0, 200 mM NaCl and 5 mM DTT. The Flag tagged VASH2_α1/α2 (94-355) was used for detyrosination activity assay without Flag cleavage.”

We have included these information in the revised manuscript.

Page 15: His tagged VASH21-66

I guess you mean SVBP1-66. Also I miss the expression conditions and how it was prepared.

Thank you for pointing out this mistake. It is His tagged SVBP₁₋₆₆. It has now been corrected in the revised manuscript.

We prepared this sample using below method:

“SVBP₁₋₆₆ was cloned into a modified pFastBac1 vector with a N-terminal His tag and was also expressed using *Spodoptera frugiperda Sf9* cells. Cells were cultured at 27 °C for 60 h after baculovirus infection, harvested and homogenized in lysis buffer containing 25 mM Tris-HCl, pH 8.0, 200 mM NaCl, 1 mM PMSF. After ultracentrifugation, the supernatant was loaded onto a Ni-NTA superflow affinity column, washed using lysis buffer plus 5 mM imidazole and eluted using a buffer containing 25 mM Tris-HCl, pH 8.0, 200 mM NaCl, 250 mM imidazole. The target was then subjected to a heparin sepharose column and finally polished using SEC 650 in a buffer containing 25 mM Tris-HCl, pH 8.0, 200 mM NaCl and 5 mM DTT.”

We have include these information in the revised manuscript.

Page 16: AUC ‘Absorbance of ...was collected at 280 nm, 250 nm and 250 nm’

I guess last absorbance value is 220 or 215 nm

You are right, and thank you for pointing out this mistake. The last absorbance value is 215 nm. We have corrected it in the revised manuscript.